# Pulses for Healthy and Sustainable Food Systems: The Effect of Origin on Market Price

**Claudio Acciani, Annalisa De Boni ***, Francesco Bozzo and Rocco Roma**

Department of Agricultural and Environmental Sciences, University of Bari Aldo Moro, 70126 Bari, Italy; claudio.acciani@uniba.it (C.A.); francesco.bozzo@uniba.it (F.B.); rocco.roma@uniba.it (R.R.)
* Correspondence: annalisa.deboni@uniba.it

**Abstract:** Pulses are widely acknowledged for their high nutritional value due to high protein content, low content in calories, and low glycemic index; they are a good alternative to animal proteins thus offering a considerable number of social, environmental, and health benefits. Despite pulses being widely acknowledged as healthy and sustainable food, in mainly European countries, consumption is growing but still lower than the recommended level, production is unprofitable in comparison to the current market prices level, and a reduction in harvested area has led to a strong dependence on import for pulses supply. Pulses are particularly fitting to the feature of local food because they can be suitably grown in any context, even in the most complex areas, and consumer interest and awareness of food origin has strongly increased in recent years. Lentils were selected as a case study in this paper that aims to define which features are effective on market price and, in particular, the role of origin declaration on label plays in defining the market price and how the origin attributes may enhance market price and farms competitiveness. The methodological tool for this investigation is the hedonic price model, useful to explain the effects of attributes of pulses affecting the market price. Results contribute to a better understanding of the pulse market, emphasizing that the "origin declaration" on label may have a positive effect on market price.

**Keywords:** pulses; origin; hedonic price; healthy food; sustainability

## 1. Introduction

Pulses are dry-harvested leguminous crops widely consumed for their very nutritious seeds, and their consumption is associated with many health benefits [1]. Pulses represent an important component of a healthy diet due to their high content of a plant protein (20–35%, on average) which can replace animal protein in the human diet, high content of carbohydrates (60%) and dietary fiber, and low in glycemic index (GI), improving diet profile in populations [2]. Their consumption have been also associated with reduced risk factors for chronic disease.

Moreover, pulses have been recognized having a crucial role in development of sustainable agriculture as their inclusion in cropping systems can enhance annual productivity by nitrogen fixation, leading to significant advantages for agricultural sustainability [3]. In fact, intercropping and rotation of pulses and cereals improve biodiversity, reducing the reliance on a cereal monoculture, and support climate change mitigation [1,3]. Pulses have also been identified as having important potential roles and significant contribution to human nutrition and food security. Legumes' nutritional and environmental value seem to be still underestimated and the level of consumption remains low [4,5] despite pulses having always been considered an important element of the human diet. In recent years, there has been increasing attention to proper nutrition, a reduction of meat consumption, and an increasing number of people following vegetarian or vegan diets.

Nearly one half of pulses' global production comes from large-scale farms with high yield levels, and it is traded internationally, although pulses are perfectly suitable in many

different agronomic conditions and may have beneficial effects on the environment. It has been noticed that pulse cultivation in small farms, with low per hectare margins, may be unprofitable in comparison to the current market prices level [5], so it becomes necessary to adopt a system of subsidies towards small farmers to improve their profitability. In order to achieve these results, both policy instruments (Common Agricultural Policy and national rural development plans measures) and specific research projects [1] have been implemented. A marketing mix based on principal characteristics positively affecting pulses market price may be crucial to enhance supply chain economic performances.

This paper aims to contribute to a better understanding of the pulses market, analyzing pulse retail prices, which are widely variable, and investigating the relationships of complementarity and/or substitution between pulses attributes. This research aims, in particular, to determine if the declaration of place of origin on label can work as a tool to improve the market price and enhance competitiveness of Italian pulses with respect to foreign or unknown origin products. Moreover, studies focused on pulses in developed countries are still scarce and mainly focused on agronomic and nutritional aspects and consumers' behavior [6–9] instead of the effect of product features on market price. This papers aims to fill this gap, supporting strategies to improve market price and profitability of pulses in developed countries strongly dependent on imports for household needs. The product considered as a case study in this paper is lentil and the specific objectives are: (a) to define which lentil features are effective on market price; (b) how the "origin" attributes may enhance market price; and (c) what role the declaration of place of origin on the label plays in defining the market price of lentils. The hedonic price model has been considered the most appropriate analytical tool for this investigation. The hedonic price model allowed distinguishing how the characteristics affect price, explaining its structure [10–13]. Lentils have been selected among other pulses because although in the 1930s Italian lentils were highly valued on foreign market, such as Germany, Canada, and Australia, their cultivation has almost disappeared. Now, the cultivation of lentils is thriving, and the consumption of the Italian product could generate a huge wealth for farmers ensuring sustainable production and national origin, but above all would bring healthy products to Italian tables [14].

## 2. Conceptual Framework

From a nutritional point of view, pulses represent a healthy source of proteins, a good alternative to animal protein consumption, thus offering, in comparison to meat, a considerable number of social, environmental, and health benefits, and may play a role in reducing meat consumption. Meat and dairy products are widely recognized as valuable sources of nutrients but, in recent years, ethical, environmental, and social reasons have inspired a critical attitude towards high levels of meat and dairy consumption [15]. As many scholars have reported [16–19], high levels of meat consumption, especially processed, are most frequently associated with an increased risk of cardiovascular disease, hypertension, and being overweight. It is acknowledged that the present ratio between plant and animal protein in the diets in industrialized countries may represent a habit that is harmful to both public health and the environment [4,19,20]. During the last decades the global consumption of pulses has remained stable at about 12 kg per capita per year, showing significant differences across countries: Latin America and the Caribbean, sub-Saharan Africa, and South Asia have the highest levels of average per capita consumption of pulses (about 12 kg per capita per year). A lower value is observed in North Africa (7.3 kg per capita per year), North America (4.4 kg per capita per year), and Europe (2.7 kg per capita per year). The recommended consumption is about 60 to 100g serving of pulses at least three to five times per week (15–25 kg per year) [5,21]. In Italy, as in other industrialized countries, per-capita consumption of dried pulses has progressively reduced from over 14 kg per year in the 1960s to the current consumption of about 1.5 kg [22,23]. These data show that consumption is still lower than the recommended level, despite pulses now being widely acknowledged as a food of high nutritional value due to their high protein

content, low content in calories, and low GI. Pulses are also a rich source of dietary fibers and complex carbohydrates and are suitable to mitigate diet-related diseases, including obesity, diabetes, cardiovascular diseases, and cancer [24].

However, in the last few years, domestic consumption of legumes has shown signs of new interest, in concurrence with an increasing consumer awareness of the health effects of the Mediterranean diet [25–27], promoting a frequent consumption of vegetables and legumes and a reduction of animal protein intake. At the same time, legumes are basic foods in the eating patterns proposed by vegetarian/vegan diets, whose popularity have increased in recent years [4,21,22]. In Italy, as in many southern Mediterranean countries, consumption of pulses is well-established and local foods are particularly well known and appreciated [28–30].

Food consumption is known as one of the most important causes of global environmental pressure. Global food production, including the entire supply chain and the effects of land-use change, accounts for about one fourth of total greenhouse gas (GHG) emissions. In recent years, several works of research focused on the evaluation of the impact of meat production [31–36] by showing the high contribution of the livestock sector to the global GHG. Legumes may be grown in any context, even in the most complex areas, such as mountain areas, hills, and marginal lands. These crops have a fertilizer role in rotations with a positive effect with respect to nitrogen when they are inserted into crop rotations and mixed crop-livestock systems. The opportunity to get short-term profitability from pulse crops in the year in which they are grown may be a promoting factor, mainly for small-medium farms. Local food supply chains with few stages between producer and final consumer are described as a means of promoting more sustainable consumption systems: legumes are suitable to be processed by farms, due to the small investment required for the processing of the seeds, including selection, drying, and packaging. It may allow farms, organized in a company or consortium, to shorten the distribution channel, retaining all the benefit of production for the farmers. It is of special interest for local ecotypes, since they have a strong territorial identity and roots. Moreover, in European Union (EU) countries, due to the European Policy (2014–2020) green direct payment, well known as "greening", there are now several actions supporting sustainable agriculture, requiring larger farm owners to diversify their production systems: legumes, with a short production cycle, may be conveniently alternated with cereals and vegetables.

In recent years, the world production of lentils has increased significantly, from 3,394,605 tons harvested in 2000 to 7,590,761 tons harvested in 2017, with an increase in the area harvested of about 60%. The global production of pulses in 2016–2018 accounted for about 77 million tons [37] showing an increase with respect to the former triennium of 15%. Dry bean production accounted for about 30 million tons, chickpea production for about 14 million tons, dry pea production for about 15 million tons, and cowpea production for about 7 million tons. The production of lentils in the same triennium was estimated to be 6 million tons.

Pulses' production is shared in every region of the world, but South Asia and sub-Saharan Africa together supply about half of global production. More specifically with regard to dry beans, Saharan Africa accounted for 24% of global production, Latin America and the Caribbean for about 24%, Southeast Asia for about 18%, and South Asia for about 17%. South Asia supplied about 74% of chickpea production. Dry peas are mainly cultivated in North America (38%) and Europe (29%). The largest production of lentils is due to North America (42%) and South Asia (30%) [38].

The largest producer of lentils is Canada, followed by India; they represent together more than 70% of the world's production [37]. The lentils are the second biggest traded pulse crop in developing countries [39] especially in Canada and Australia, where lentils are cultivated on large-scale, capital intensive farms, with high yields, and mainly (77% for Canada and 82% for Australia) allocated to Asian and European countries. The static or declining production trends in developing countries, despite increasing global demand, threatens current and future food security [3,40].

Although Italian pulse production is just 0.2% of the global total, the role of Italy is not marginal in the European context, as the Italian production of pulses represents 5% of the EU's total production [22]. Italian pulse production declined starting in the 1930s from about 800,000 tons to the current 150,000–200,000 tons, due to a simultaneous decrease in cultivated areas and an improvement in the average yields, which made the loss of production less remarkable. Moreover, Italian pulse production has also decreased because of the massive import of dry pulses of low quality and low cost from non-EU countries (accentuating foreign dependence especially the United States (U.S.A.), Mexico, the Middle East, and Turkey). This phenomenon damages not only Italian pulse producers but also the environment and citizens, because the foreign regulatory framework about the use of chemicals, as in the case of glyphosate and as regards working conditions, is much less restrictive than in Europe. This makes it possible to produce at very low costs, dumping on farmers in Italy and putting the health of consumers at risk. In particular, lentils and beans are estimated to be of extra-community origin for 90% of total consumption, 70% of peas are imported, and about half of chickpeas consumed is produced in non-EU countries [14]. In particular, canned pulses may be imported as dry pulses, rehydrated, and then put in a can as Italian product. Legumes that explicitly highlight the national origin in the label or those sold directly by farmers prevent consumers from falling into the deception of the fake "Made in Italy" buying an imported dry product.

As regards lentils, a strong regional specialization was observed, leading to a concentration of the cultivated area for each species in only a few regions, so that, at present, about 88% of Italian production is located in the first 5 producing regions [41]: Umbria (25%), Tuscany (23%), Marche (14%), Apulia (12%), and Sicily (6%). With respect to beans, lentils have a greater genetic variability, reduced water and nutritional requirements, and, consequently, a higher potential for environmental adaptability. Moreover, lentil cultivation was almost abandoned in the last decades but has shown in recent years' strong signs of recovery, regarding both area harvested and production. The growing interest in this crop is also demonstrated by the attribution of a new designation of origin ("Altamura Lentil IGP") in 2017. Local lentil populations are still cultivated in these regions and their large genetic variability makes them suitable for low input or organic farming management, but local populations are strongly threatened by genetic erosion and loss of biodiversity, due to the substitution with more productive foreign cultivars. More specifically, consumption of legumes could contribute to defining more sustainable consumption systems, stimulating the development of new processing and marketing activities and promoting health, local employment, and environmental sustainability. Valorization of legume crops through their link with the territory may represent a new economic opportunity [42], and, not least, it is perfectly consistent with the 2nd Sustainable Development Goal (SDG), "Zero Hunger", because it may harmonize healthy food production and sustainable agriculture.

Italy is the EU country that can boast the highest number of protected designation of origin (PDO) and protected geographical indication (PGI) products. Currently, 299 Italian food products and 523 Italian wines are recognized as PDO or PGI. In 2019, the production turnover for Italian food covered by a designation of origin was about 6.96 billion €, the sales turnover was 14.7 billion €, and export was more than 3.5 billion €. More than 83,000 operators are involved in the designation of origin products supply chain. In Italy, there are two lentils worthy of indication of geographic protection (IGP): the lentils "Altamura Lentil IGP" and "Castelluccio Lentil IGP".

In Apulia, there were about 10,000 hectares of lentils cultivated in 2018 [41]: 2,000 hectares are certified for the production of "Altamura Lentil" that was awarded the IGP label in 2017, about 1400 tons were harvested in the 2017/2018 agricultural year, with an average of 700 kg per hectare. In Umbria, about 400 tons of "Lenticchia di Castelluccio IGP" were harvested and cleaned during the months of August and September 2018 on about 500 certified hectares, with an average of about 800 kg per hectare.

In recent years, a growing consumer interest in food origin has been observed and numerous international studies stressed consumers' willingness to pay a premium price

for domestic and local food [43–45]. Likewise, many studies demonstrated a significant premium price for products labeled as locally produced [12,46].

Many scholars [47–50] underlined an increasing interest among consumers toward local food products, perceived to be of higher quality according to their features of healthiness (fresher, tastier, healthier, safer, etc.); ecological sustainability (due to the use of sustainable production methods and minimal transport); and social sustainability (supporting local economies and community). The concept of local food, although it is abundantly used [43,51–54] to describe local food systems or short food chains where food is produced near the consumer or from within their own region, is not, however, well-defined.

Some scholars [55–58] underlined that Italian consumers consider the place of production and processing of food as factors of highest importance in determining food quality. Regardless of their definitions, "local" foods are perceived as valuable products because of a wide range of benefits [59,60]. Local foods are perceived as having the capability to improve the environmental and economic sustainability of food supply chain: they have a positive effect on the carbon footprint due to the low transport carbon emission [27,32]; they offer new market opportunities for local farms thus supporting the local economy and playing a strategic role in promoting sustainable territorial management and rural development because of their ability to satisfy consumers' demand with food from local and natural resources; preserve traditional food culture; and increase local employment [52,61].

The lack of an official definition and regulation of "local food" through standardized labels may make it difficult for consumers to identify local products. Frequently, the definition of local food is referred to in distances (i.e., miles or kilometers) or to the driving time: according to some scholars a radius of 30 or 50 miles is associated with local food, whereas a 100-mile radius is generally referred to food that is considered more regional than local [13,62,63]. Other times the definition of "local" arises from a set of emotional and social features that are strongly influenced by contextual factors such as knowledge and involvement with the food and social context, pushing consumers' choices toward home-grown food or towards food produced by neighbors, relatives, and friends [64,65]. In this case local product purchases have a positive role in supporting local producers, especially those operating small or family-owned enterprises, and a positive impact local economy and employment. Furthermore, consumers feel that purchasing local products allows their money to remain within their own community to support [13,66]. Generally, the trade of local products is implemented through short chains with few or no intermediaries and often based on personal interaction between the producer. In other cases, the local food identity is related to a specific district or administrative area, inside political boundaries. Some scholars consider "local" a food produced within the same county [62], whereas others consider larger frontiers, such as the state [46], and even the nation [13,52,59], and moreover, based on specific speciality features and brand names associated with a region [67]. According to other authors [68], regional products having an origin from a particular locality or region can be considered local products on a larger scale because they are sold in large external markets and sometimes referred to "locality foods" to distinguish from local foods. Locality foods are identified as having been produced and processed in a particular place but often circulate more widely (e.g., Orkney Cheddar and Arbroath Smokies) [69–71].

## 3. Materials and Methods

### 3.1. Hedonic Price Methodology

According to the hedonic price approach, consumer utility is not generated by the purchased product itself, but rather by the qualities and characteristics it contains; the goods characteristics and not the goods in itself give rise to consumer utility [72]. Differentiated goods may be considered as a set of various quality attributes discriminating them from other similar goods, so that the equilibrium market price may be considered a function of the implicit prices of each good attribute [73]. The prices drive both consumer and producer choices. In particular, producers tailor their goods to satisfy final characteristics

desired by customers [69], but the price functions should not be directly interpreted as general measures of consumer willingness to pay for product attributes: the high implicit prices observed for an attribute may be more affected by elevated costs of production, than by a high appreciation from consumers, and it is possible that only a small fraction of consumers actually purchase goods containing expensive attributes.

Nevertheless, the hedonic approach is generally acknowledged as a good tool to evaluate the premium price for "credence attributes" such as certification, indications of origin, and other characteristics that consumers cannot verify otherwise, even after food purchase and consumption [73,74]. Though the model of the hedonic price is widely used to determine implicit prices in food and beverage sector, most recent studies are focused on wine, [75–77] oil [78], diary, eggs and meat [79–83], and fish [84,85]. To the best of our knowledge, no studies have explicitly analyzed pulses using hedonic price functions. In the context of a relatively unexplored market, as is the case for pulses, the hedonic price model may give the pulse producer useful insights into the most important attributes to which they must pay attention to improve their profit.

The hedonic price model allows analyzing the relationship between the price and the main quality attributes of a product. Goods are described by n objective features, and their hedonic price function, in its simplest form, may be expressed as: $p(z) = P(Z_1, Z_2, \ldots, Z_n)$ where $Z_i$ assesses the amount of the ith characteristic contained in each goods [83].

Since variables with different scales as well as dummy variables may be selected, it is necessary to find the optimal hedonic pricing model among the available forms, such as log-linear, double log-linear, and linear log. In this study, a single-equation approach was applied [80,81] to determine the effects each feature of lentils causes on price. A semi-logarithmic specification of the hedonic price equation was preferred using the natural logarithms of the dependent variable (average price per kg) and weight; the log-linear specification for a hedonic pricing model was thought to be most suitable for the data according to several tests.

The final model specification and equation is given as:

$$\ln P = \beta_0 + \beta_j \ln Z_i + \beta_j X_{ij} + \varepsilon \tag{1}$$

where P is the price, $\beta_0$ is the constant, $\beta_j$ is a vector of parameters of product attributes; Z is the quantitative attribute "weight", X is defined as a set of observable product qualitative attributes, subscript i identifies an attribute (i = 1, $\ldots$ , I); j is the number of choices across different qualitative attributes (j = 1, $\ldots$ , J); $\varepsilon$ is residual, and ln is the natural log.

### 3.2. Materials

After a desk analysis on scientific literature and other commercial papers, and a focus group developed in January 2020 involving academic researchers mainly devoted to marketing studies, stakeholders, and sector operators (4 producers, 4 retailers, 1 consortium of Altamura lentils IGP representative, and 2 academic researchers), 8 lentil characteristics were defined. Some characteristics were intrinsic lentil features: seed color and size, organically or conventionally cultivated, origin (if regional or sub-regional area of origin is declared on label), and geographical indications of origin (IGP); other features were extrinsic: such as packaging and weight, retail store type (large scale distribution (LSD), specialized organic retail, discount, or grocery). Lentil price and characteristic data were collected directly in 38 outlets in metropolitan areas; outlets were randomly selected from a list of all retail stores obtained from a current local online phone directory, in February 2020. Data were collected by researchers involved in this research, after receiving the authorization from the person responsible for each retail. A survey form, containing for each package of lentils on the shelves all the variables reported in the first column of Table 1, was filled. Collected data were reported in a database and submitted and analyzed through IBM SPSS21 software.

**Table 1.** Variable description.

| Description | Variables | Category |
|---|---|---|
| | Dependent variable: | |
| Price | Sale price €/package | Continuous |
| | Explanatory variables: | |
| Package size | Package content in grams | Continuous |
| Package kind | Cardboard box = 1; otherwise = 0 | Dichotomous |
| | Bagful = 1; otherwise = 0 | Dichotomous |
| | Unpackaged = 1; otherwise = 0 | Dichotomous |
| Organic certification | Certified = 1; not certified = 0 | Dichotomous |
| Geographical indications of origin: protected designation of origin (PDO) indication of geographic protection (IGP) | Certified = 1; not certified = 0 | Dichotomous |
| Origin | Regional or sub-regional origin on label = 1; otherwise = 0 | Dichotomous |
| Seed size | Small = 1; otherwise = 0 | Dichotomous |
| | Medium = 1; otherwise = 0 | Dichotomous |
| | Large = 1; otherwise = 0 | Dichotomous |
| Seed color | Red = 1; otherwise = 0 | Dichotomous |
| Store | Large Scale Distribution (LSD) = 1; otherwise = 0 | Dichotomous |
| | Specialized = 1; otherwise = 0 | Dichotomous |
| | Discount = 1 otherwise = 0 | Dichotomous |
| | Organic shop = 1 otherwise = 0 | Dichotomous |

Product prices obtained from retailers were measured in euros per package. The package price was divided by the weight of package and referred to one kilo of product. Regarding qualitative variables, they were considered dichotomous variables and coded as 1 if present and 0 if otherwise. Variables and categories descriptions are summarized in Table 1.

*3.3. Empirical Specification*

The empirical hedonic pricing equation applied was the following:

$$\ln P = \beta_0 + \beta_1 \ln X_1 + \beta_2 X_2 + \beta_3 X_3 + \beta_4 X_4 + \beta_5 X_5 + \beta_6 X_6 + \beta_7 X_7 + \beta_8 X_8 + \varepsilon \qquad (2)$$

where P is the price, $X_1$ is the package size, $X_2$ is the pack type, $X_3$ is the organic certification, $X_4$ is the geographical indications of origin, $X_5$ is the origin, $X_6$ is the seed size, $X_7$ is the seed color, and $X_8$ is the store; $\beta_0$ is constant, $\varepsilon$ is residual, and ln is the natural log.

This allowed empirically determining the marginal implicit prices for each attribute. Variables description and categories are summarized in Table 1. Regarding qualitative variables, they were considered dichotomous variables and coded as 1 if present and 0 if otherwise.

**4. Results**

The collected data allowed to perform the sample descriptive statistics, summarized in Table 2, revealing that the average lentils price was 4.7 €/kg, with a high variability from the maximum price 17.9 €/kg, observed for 2.5% of cases characterized by the smaller pack size (250 g) and the presence of PDO label, and the minimum price of 1.8 €/kg, for 2.5% of cases too, related to conventional and small size lentils, in 500 g bags, sold in discount. Results from ANOVA showed that package size had a significant (F = 25.2 and sign 0.000) influence on price, in accordance with the higher prices observed for smaller lentils packages; conventional lentils had a price significantly lower than organically certified (F = 18.4; sign = 0.000). Regarding origin, lentils endowed with geographical indications of origin (PDO/IGP) (F = 9.1; sign = 0.003) or labeled with a regional or sub-

regional indications, showed a market price significantly (F = 14.3; sign = 0.000) higher than lentils without any certification of origin. Finally, lentils sold in discount stores showed a significantly (F = 14.3; sign = 0.000) lower price than those purchased in other stores (LSD; specialized and organic store).

**Table 2.** Descriptive statistics.

| Variables | Observation (%) | Prices (€/kg) | | | S.D. | F | Sign | |
|---|---|---|---|---|---|---|---|---|
| | | Min. | Max. | Average | | | | |
| | | | Total sample | | | | | |
| | 100 | 1.8 | 17.9 | 4.7 | 3.1 | | | |
| | | | Package size | | | | | |
| 250 g | 2.5 | 17.9 | 17.9 | 17.9 | 0.0 | 25.2 | 0.000 | ** |
| 350 g | 2.5 | 7.7 | 8.5 | 8.1 | 0.6 | | | |
| 400 g | 28.4 | 3.2 | 9.3 | 5.6 | 2.1 | | | |
| 500 g | 55.6 | 1.8 | 9.5 | 3.9 | 2.0 | | | |
| 1000 g | 1.2 | 1.9 | 3.0 | 2.5 | 0.5 | | | |
| | | | Package kind | | | | | |
| Cardboard | 14.8 | 2.4 | 18.0 | 6.1 | 4.6 | 3.0 | 0.088 | |
| Bagful | 75.3 | 2.0 | 18.0 | 4.7 | 2.8 | 0.0 | 0.999 | |
| Unpackaged | 9.9 | 2.0 | 3.0 | 2.6 | 0.5 | 4.3 | 0.041 | |
| | | | Organic certification | | | | | |
| Conventional | 71.6 | 2.0 | 18.0 | 3.8 | 3.1 | 18.4 | 0.000 | ** |
| Organic | 28.4 | 4.0 | 10.0 | 6.8 | 1.8 | | | |
| | | | Seed size | | | | | |
| Small | 31.7 | 2.0 | 18.0 | 4.6 | 3.0 | 0.0 | 0.856 | |
| Medium | 12.3 | 2.2 | 10.0 | 4.8 | 2.6 | 0.0 | 0.881 | |
| Large | 25.9 | 2.0 | 18.0 | 4.7 | 3.6 | 0.0 | 0.930 | |
| | | | Seed color | | | | | |
| Red | 23.5 | 2.0 | 8.0 | 3.7 | 1.5 | 2.5 | 0.116 | |
| Green | 76.5 | 2.0 | 18.0 | 5.0 | 3.4 | | | |
| | | Geographical indications of origin (PDO/IGP) | | | | | | |
| Certified | 11.1 | 3.0 | 18.0 | 7.4 | 6.0 | 9.1 | 0.003 | |
| Not certified | 88.9 | 2.0 | 10.0 | 4.3 | 2.3 | | | |
| | | | Store | | | | | |
| LSD | 58.0 | 1.9 | 17.9 | 5.1 | 3.4 | 2.7 | 0.103 | |
| Specialized | 9.9 | 2.2 | 9.3 | 5.0 | 2.5 | 0.1 | 0.721 | |
| Discount | 23.5 | 1.9 | 5.6 | 2.5 | 0.9 | 14.3 | 0.000 | ** |
| Organic shop | 8.6 | 5.0 | 8.9 | 6.9 | 1.2 | 4.3 | 0.040 | |
| | | | Origin | | | | | |
| Regional/sub-regional | 24.7 | 3.0 | 18.0 | 7.5 | 4.1 | 31.0 | 0.000 | ** |
| Italy or foreign | 75.3 | 2.0 | 9.3 | 3.7 | 1.9 | | | |

* Significant at α = 0.05; ** significant at α = 0.01.

The obtained database, including 162 observations and 15 explaining variables, was submitted, in advance, to a correlation analysis that highlighted lower correlation index (r max 0.62), and later to a linear forward stepwise regression, following the rule that there be at least five observations for each factor [86,87]. The empirical model showed a good overall significance (F statistic equal to 19.502 with a *p* value much lower than 0.0000) and a good capability to explain the variability of the dataset ($R^2$ = 0.534) (Table 3). Furthermore, four of the estimated coefficients considered in this model were statistically significant

(Table 3) and significantly affected the price estimates of the hedonic model. Results in the parameter values are summarized in the same table.

**Table 3.** Parameter estimates for hedonic model.

| Variables | Coefficient β | t | Sig. | |
|---|---|---|---|---|
| Constant | 9.757 | 4.140 | 0.000 | ** |
| Store (discount) | −0.415 | −3.281 | 0.002 | ** |
| Organic certification | 0.254 | 2.287 | 0.025 | ** |
| Regional/sub-regional origin | 0.263 | 2.071 | 0.042 | * |
| Package size (ln weight) | −1.374 | −3.603 | 0.001 | ** |
| $R^2$ | 0.534 | | | |
| Adj.$R^2$ | 0.507 | | | |
| F-statistic | 19.502 | | | |

* Significant at $\alpha = 0.05$; ** significant at $\alpha = 0.01$.

The empirical model may now be expressed according to the following expression:

$$\ln \text{price } (\text{€/kg}) = 9.757 - 0.415 \times \text{discount} + 0.254 \times \text{organic certification} + 0.263 \times \text{regional/sub-regional origin} - 1.374 \times \text{package size (ln weight)} \tag{3}$$

In Figure 1, prices calculated using Equation (3) are reported. Each price value was calculated by replacing in Equation (3) the value of each significant variable. In particular, for the baseline (dark gray column), characterized by the most frequent features (conventional lentils, small- or medium-sized, with no origin-related indications, sold in LSD in a 500 g bag), the variables "discount", "organic certification" and "regional/sub regional origin" assumed value "0", while for the weight the natural logarithm of 500 was calculated. In this case, an average price of 3.4 €/kg resulted. Other price valued were calculated respectively, by assigning value 1 to the variables "discount" (discount = price 2.2 €/kg), 600 to the variable weight (600 g = price 2.6 €/kg), 1 to the variable "organic certification" (organic = price 4.4 €/kg), 1 to the variable "regional/sub-regional" (regional/sub-regional = price 4.5 €/kg). Discount and a bigger size of packaging decreased price; organic and origin declaration increased it.

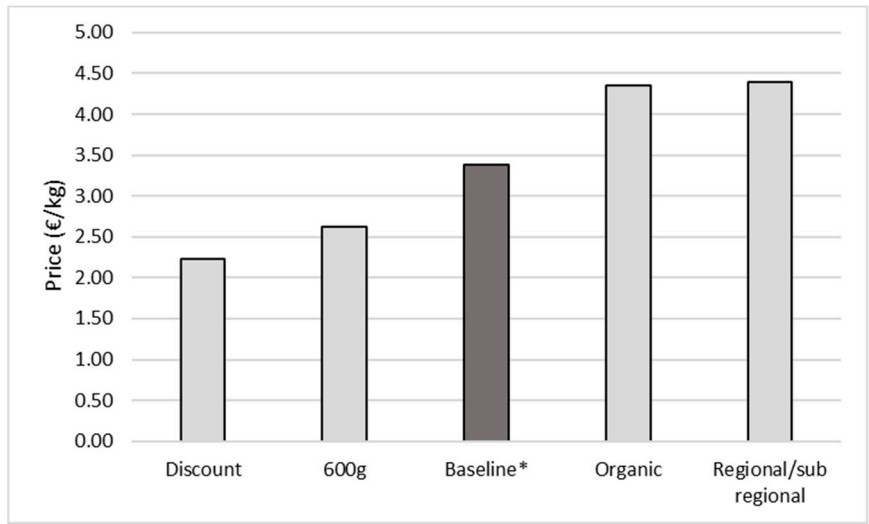

**Figure 1.** Lentil prices and characteristics in hedonic model. * Baseline represents conventional lentils, of small or medium size, with no origin-related indications, sold in large scale distribution (LSD) retail stores, in a 500 g bag.

## 5. Discussion

As regards the intrinsic characteristics, the results clearly showed that the seed size and color did not statistically influence the price. Nevertheless, concerning the intrinsic variable "size", other studies [24,39] highlighted a consumer preference for larger lentils, which are considered tastier once cooked and time saving for cleaning prior to cooking. Lentils, in comparison to other pulses (beans, chickpeas) do not require a long time for cooking; this may suggest that, for lentils, the limited time available for food cooking is not considered a hindrance to healthy food choices [88]. "Regional/sub-regional" features had the highest positive ($\beta$ = 0.263) and statistically significant effect on the product's price too: in terms of monetary value, the origin attribute added a premium price of 1.0 €/kg (+30%) respective to the baseline lentils price (Figure 1).

Insights from market price analysis confirmed that the "origin" attribute of lentils is related not only to their geographical origin, but also to tradition and culture, natural factors, and the use of ancient seeds [10,54,80,89]. Moreover, consumers tend to attribute a "typicality" to food on the basis of place of origin, traditional production, and processing techniques. Foods having a special link with a given territory can be considered the result of natural and cultural resources of a specific geographic area. Foods whose production process is linked with specific localities may be identified with qualitative excellence so that consumers recognize a value added for original ingredients and taste and may express a significant willingness to pay a premium price for local and origin certified foods [60]. The observed price surplus is consistent with the findings in [90]: purchases of food with a particular origin usually represent a small proportion of food expenditures and so it is slightly affected by income and slightly affects total food expenditures; moreover, lentils are widely considered as a commodity and a low cost alternative to more expensive animal-based protein sources (meat or cheese). As a matter of fact, for consumers, the origin is a more important concern than the attributes of lentils such as visual appearance, texture and flavor, especially when lentils are sold in bulk [91]. Some authors [92] highlighted that bulk selling may inspire the feeling of a closeness to the product's naturalness, thus reducing the perceived environmental impact of the products [93–97] and may be considered appealing and reminiscent of "farming", and, once again, reminiscent of the local origin. Hence, consumers might attribute to regional or sub-regional foods the same quality of products protected by geographical indications, which can be viewed as legal signals of a certain provenance and local tradition [98,99]. In comparison to the industrialized and standardized food products, products of defined origin are able to strengthen consumer confidence; product origin is growing in importance in standardized and globalized markets as a reassertion of regional culture and a demand for individuality and authenticity [100].

Our results suggest that market price may be significantly influenced by the exogenous effect of the origin-declaration on products. That is in line with other studies [55–60,82] stressing the positive attitude of Italian consumers to recognizing a higher value to domestic and local products. That points to possible changes in market practices. The adoption of voluntary labeling options pointing out the origin from a specific territory may be effective and can help producers and sellers to differentiate the Italian product from the imported one.

In fact, the premium price imputable to organic certification is very similar to the premium price due to the local attribute. Organic certification positively affected the price too ($\beta$ = 0.254): a premium of 0.97 €/kg (+28.8%) could be achieved if the product was organically certified and this result is consistent with the insights of many other authors [43,101,102]. On the supply side, it is worth noting that the premium prices associated with organic production does not reflect higher production costs: the cultivation of organic lentil request less fertilizer, pesticide, and labor cost respective to conventional lentils, and may benefit from subsidies from the measures for organic cultivation development present in the regional rural development plan (RDP). A possible reason is that rapid expansion of the organic farming and the increased availability on the market of organic lentils has reduced the gap with conventional products enlarging the number of organic consumers. Moreover, it highlights consumers' attitude to acknowledging a higher value to organic

food. This result may be explained taking into account the growing consumer attention to health and environmental sustainability, which are becoming important food choice criteria.

The variables that negatively affected the price were the package size (β = −1.374) and store kind (β = −0.415). Price reduction for larger packages, which implies economies of scale, is consistent with other authors' findings [74,103]. An increase of 100 g in bag weight caused a decrease of 22.2% in the final price, consistent with the practice of weight discounting that retailers may offer with packaged food [81,102], in particular for products such as dried legumes with a long shelf-life, avoiding food waste [104]. It is expected that a lower price may encourage consumers to stock up by purchasing largest package.

Lentils sold in a discount grocery retail showed a price discount of 33.9%, compared to the baseline option (Figure 1). It is not surprisingly and consistent with other authors' findings [105] that discount stores are able to keep costs down applying a very simple store management, in which products are often displayed on the floor on pallets and retail-ready, and the number of employees is as small as possible. In the end, discount grocery retailers might contribute to the increase of consumption of healthy products and more sustainable diets even with regard to more price conscious consumers [106]. A possible strategy to increase family consumption of local pulses, answering the need to support healthier and adherent to the patterns of the Mediterranean diet food consumptions [25,27,30], could be boosting local food presence in large scale distribution and discount outlets and in alternative food purchasing initiatives, such as box schemes or solidarity purchasing groups [64]. Suggestions to companies result from our insights include: investing in voluntary labeling actions supplying consumers with a clear indication of place of origin and explaining the environmental and social benefits of local pulses; and fruiting the synergy that can result from the indication of origin combined with organic certification [64]. Moreover, policy interventions supporting valorization strategies of legume crops through their link with the territory may respond to the 2nd Sustainable Development Goal (SDG) "Zero Hunger", harmonizing healthy food production and sustainable agriculture.

## 6. Conclusions

The applied hedonic price model was effective with the prior purpose of analyzing all the lentil's features affecting market price. In contrast to methods based on the analysis of consumer behavior, the hedonic price method, recording the actual market prices, allows to overcome the uncertainty related to the difference between what the consumer states and what will be his actual purchasing behavior. As expected, for lentils, because of their nature as a staple food, credence attributes as organic certification and origin declaration influenced market price more than other intrinsic characteristics like seeds size or color. The results of this research confirm that the place of production is a relevant aspect in determining lentils' market price. Among limitations of this study it is to be underlined that data collection was performed mainly in southern Italy, so the results may be carefully applied to the rest of Italy. Nevertheless, the collection of data for the hedonic price analysis was completed in all types of sale outlets and a strong uniformity of products on shelves emerged. This is a confirmation of the commodity feature of lentils and of the effectiveness of extrinsic features to explain market price variability. Despite these limitations it should be pointed out that this research insight could be useful for the suggestion of marketing strategies and policies supporting national legumes sector. The results call for good market opportunities for local productions, both awarded with a designation of origin and declaring their origin on the label, especially if they will be purposely promoted by underlining their special links to the production areas. The findings of this study may facilitate farmers, traders, and policy makers in strategically promoting lentils, and pulses in general, with specific attributes of origin. Therefore, actions increasing consumer awareness about pulses' health benefits and role in improving sustainability of the entire food system are still required to increase the public perception of legumes as a key element for food security.

**Author Contributions:** Conceptualization: A.D.B., R.R. and C.A.; methodology: A.D.B., R.R. and C.A.; software use: A.D.B., R.R. and C.A.; validation: A.D.B., R.R. and C.A.; formal analysis: A.D.B., R.R. and C.A.; investigation: A.D.B., R.R. and C.A.; data curation: A.D.B., R.R., C.A. and F.B.; writing—original draft preparation: A.D.B., R.R. and C.A.; writing—review and editing: A.D.B., R.R. and F.B. All authors have read and agreed to the published version of the manuscript.

**Funding:** This research received no external funding.

**Institutional Review Board Statement:** Not applicable for studies not involving humans or animals.

**Informed Consent Statement:** Not applicable for studies not involving humans.

**Data Availability Statement:** The data presented in this study are available on request from the corresponding author. The data are not publicly available due to the type of authorization for their collection by store managers.

**Acknowledgments:** The authors would like to thank Paolo Ceci Ginistrelli for providing support in data collection and database implementation.

**Conflicts of Interest:** The authors declare no conflict of interest.

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
