# Peer review of "Pulses for Healthy and Sustainable Food Systems: The Effect of Origin on Market Price"

_sustainability, doi:10.3390/su13010185_

Round 1

Reviewer 1 Report

Why selected lentils vs beans to analysis? Please justified the selection.

How is collected the data? Explain it please. How Is present the survey and what part contained.

Table 2 show the results but, is the final model? The rest of variables were included in others models? Is the cross effect of the variables to be into account in some estimation?

Please explain better how is obtain the results of the figure 1

The conclusion could be include the policy recommendation for firm and government from the results obtained.

Author Response

Thank you for your useful suggestions. We considered all your comments and emended the manuscript. (text in green)

Why selected lentils vs beans to analysis? Please justified the selection.

Thank you for your suggestion. We added explanations in introductive section (line 67-71) and in conceptual framework (line 166-171)

 Line 67-71 Lentils have been selected among other pulses because although in 1930s Italian lentil was highly valued on foreign market, such as Germany, Canada and Australia, then their cultivation has almost disappeared. Now the cultivation of lentils is thriving and the consumption of the Italian product could generate a huge wealth for farmers ensuring sustainable production and national origin, but above all would bring healthy products on Italian tables [14].

Line 160-165 . Respect to the beans the lentil has a greater genetic variability, reduced water and nutritional requirements and, consequently, a higher potential for environmental adaptability. Moreover lentils cultivation was almost abandoned in the lasts decades but showed in recent years’ strong signs of recovery both regarding area harvested and regarding production. The growing interest in this crop is also demonstrated by the attribution of a new designation of origin (“Altamura Lentil IGP”) in 2017..

How is collected the data? Explain it please. How Is present the survey and what part contained.

Thank you for your suggestion. A explanation about data collection procedure has been added.

See lines 281-284.

Data were collected by researchers involved in this research, after receiving the authorization from the person responsible for each retail. A survey form, containing for each package of lentils on the shelves, all the variables reported in the first column of table 1, was filled. Collected data were reported in a data base and submitted analysed through IBM SPSS21 software.

Table 2 show the results but, is the final model? The rest of variables were included in others models? Is the cross effect of the variables to be into account in some estimation?

Thank you for your comment.

Final empirical model is the one reported in equation (3); we started from the correlation matrix analysis showing that there were no correlated variables (r max 0.65); then, as specified in the text, (line 316-317) the variables were reported and the regression procedure eliminated those that showed a low probability of influencing the price, which is the reason for our research; moreover, no cross effect indicators were calculated by the SPSS procedure used. Therefore, alternative models have not been studied even if it could be interesting to deepen them in other researches

Please explain better how is obtain the results of the figure 1  

Thank you for your comment. We added a more detailed explanation from line 330 to 338

Each price value has been calculated by replacing in the equation (3) the value of each significant variable. In particular, for the baseline (dark greyed column), characterized by the most frequent features, (conventional lentils, small or medium sized, with no origin-related indications, sold in LSD in a 500g bag) the variables "discount", "organic certification" and "Regional/sub regional origin" assumed value "0", while for the weight the natural logarithm of 500 has been calculated. In this case an average price of 3.4€/kg resulted. Others price value were calculated respectively, by assigning value 1 to the variables "discount" (discount=price 2.2€/kg), 600 to the variable weight (600g= price 2.6 €/kg), 1 to the variable "organic certification" (organic=price 4.4€/kg), 1 to the variable “Regional/sub regional” (Regional/sub regional=price 4.5€/kg).

The conclusion could be include the policy recommendation for firm and government from the results obtained.

We added some comments from line 409 to 419

A possible strategy to increase family consumption of local pulses, answering the need to support healthier and adherent to the patterns of the Mediterranean diet food consumptions [26,28,31] could be boosting local food presence in large scale distribution and discount outlet and in alternative food purchasing initiatives, such as box schemes or solidarity purchasing groups [65]. Suggestions to companies result from our insights: investing in voluntary labelling actions supplying consumers with a clear indication of place of origin and explaining the environmental and social benefits of local pulses; fruiting the synergy that can result from the indication of origin combined with organic certification [65]. Moreover, policy interventions supporting valorisation strategies of legume crops through their link with the territory may respond to the 2nd Sustainable Development Goals (SDGs) "Zero Hunger", harmonizing healthy food production and sustainable agriculture.

Best regards

Reviewer 2 Report

Dear Author(s),

This is an interesting paper on a topical research area. While the topic is clearly aligned with our journal’s aim and scope, there are several concerns regarding the practical relevance and conceptual foundation for your study, as well as the research design, that make it difficult to determine if or how your study makes a significant contribution to previous research in this area.

It is recommended that the authors address the following comments and suggestions.

ABSTRACT

Please add a clear research objective in the abstract. As this should be the starting sentence for the abstract, you have to try to sharpen its focus.

Overall, I would make the abstract sharper, answering to the following questions:

- What is the problem?

- What have you done?

- What is the main contribution of the paper?

INTRODUCTION

The motivation for this study in the introduction is not strong enough. Be clearer about why we should care about this topic and what the gap is in the literature.

My chief concern is that the originality of the paper is not clearly explained nor in the abstract, nor in the introduction.

LETERATURE REVIEW

Your introduction and literature review should be like a story to read. This means that they should be interrelated: the introduction presents the main topics, then they have to be explored in deep in the literature review. This is not the case.

Moreover, in the whole paper, the authors used some old citations and it is suggested to read the recent papers and add citations.

METHODOLOGY

Methodology section mainly is the paper’s argument built on an appropriate base of theory, concepts, or other ideas. The method and methodology employed should be explained and correctly interpreted taking into account the reason for choosing the current methodology and adding information about past studies that applied the same methods in similar areas. Please, pay attention to these issues.

DISCUSSION

There is a need for more discussion on how achieved findings can be connected to the previous conceptual background. Please link the discussion of the research problem with the highlighted gaps in the conceptual background. In fact, it is expected to use and criticise any relevant literature (in the literature review section) in order to reach the main purpose of the study to explain how the paper succeeded in enriching the development of the selected topic as well as to explore different views.

There are some typos, please make sure you proofread the paper

THEORETICAL AND PRACTICAL IMPLICATIONS

To begin, the justification for your study from an industry/practical standpoint was lacking. This was a salient oversight on your part.

You should try to answer to the following questions: What kinds of objective evidence can you offer that would make industry leaders sit up and pay attention to your study? What makes this topic a big deal right now, and perhaps in the immediate future?

Best Regards

Author Response

Thank you for your suggestion. The abstract has been revised according your advices. Our amendment according your comments are reported in red in manuscript

Abstract: Pulses are widely acknowledged of their high nutritional value due to high protein content, low content in calories and glycemic index; they are a good alternative to animal proteins thus offering a considerable number of social, environmental and health benefits. Despite pulses are widely acknowledged as healthy and sustainable food, in mainly European Countries, consumption is growing but still lower than the recommended level, production is unprofitable in comparison to the current market prices level and a reduction in harvested area has led to a strong dependence on import for pulses supply. Pulses are particularly fitting to the feature of local food because they can be suitably grown in any context, even in the most complex areas and consumer interest and awareness of food origin has strongly increased in recent years. Lentils were selected as a case study in this paper that aims to define which features are effective on market price and, in particular, the role of origin declaration on label plays in defining the market price and how the origin attributes may enhance market price and farms competitiveness. The methodological tool for this investigation is the hedonic price model, useful to explain the effects of attributes of pulses affecting the market price. Results contribute to a better understanding of pulse market, emphasizing that the “origin declaration” on label may have a positive effect on market price.

Please add a clear research objective in the abstract. As this should be the starting sentence for the abstract, you have to try to sharpen its focus.

Overall, I would make the abstract sharper, answering to the following questions:

- What is the problem?

We added in the abstract (lines 12-16) the sentences:

 Despite pulses are widely acknowledged as healthy and sustainable food, in mainly European Countries, consumption is growing but still lower than the recommended level, production is unprofitable in comparison to the current market prices level and a reduction in harvested area has led to a strong dependence on import for pulses supply. –

What have you done?

.Abstact (lines 18-23)

 Lentils were selected as a case study in this paper that aims to define which features are effective on market price and, in particular, the role of origin declaration on label plays in defining the market price and how the origin attributes may enhance market price and farms competitiveness. The methodological tool for this investigation is the hedonic price model, useful to explain the effects of attributes of pulses affecting the market price.

 What is the main contribution of the paper?

Abstact lines 23-24.

Results contribute to a better understanding of pulse market, emphasizing that the “origin declaration” on label may have a positive effect on market price.

INTRODUCTION

The motivation for this study in the introduction is not strong enough. Be clearer about why we should care about this topic and what the gap is in the literature.

My chief concern is that the originality of the paper is not clearly explained nor in the abstract, nor in the introduction.

Thank you for your comment. More explanations, supported by new and more recent literature, have been added in the introductive section

Line 54-61.

This paper aims to contribute to a better understanding of pulses market, analysing pulses retail prices, which are widely variable, and investigating the relationships of complementarity and/or substitution between pulses attributes. This research aims, in particular, to determine if the declaration of place of origin on label can work as a tool to improve the market price and enhance competitiveness of Italian pulses respect to foreign or unknown origin products. Moreover, studies focused on pulses in developed countries are still scarce and mainly focused on agronomic and nutritional aspect and consumers’ behaviour [6-9] instead of the effect of product features on market price. This papers aims to fill this gap, supporting strategies to improve market price and profitability of pulses in developed countries strongly dependent on imports for household needs.

LETERATURE REVIEW

Your introduction and literature review should be like a story to read. This means that they should be interrelated: the introduction presents the main topics, then they have to be explored in deep in the literature review. This is not the case.

Thank you for your suggestion. Introduction and Literature review have been reorganized and made consistent according three main topics: effects on health of pulses consumption, environmental effects of pulses consumption and cultivation, market and trade dynamics and preferences for local pulses. New recent references have been added.

Moreover, in the whole paper, the authors used some old citations and it is suggested to read the recent papers and add citations.

 We added new references (in red in references list)

METHODOLOGY

Methodology section mainly is the paper’s argument built on an appropriate base of theory, concepts, or other ideas. The method and methodology employed should be explained and correctly interpreted taking into account the reason for choosing the current methodology and adding information about past studies that applied the same methods in similar areas. Please, pay attention to these issues.

Thank you for your suggestion, we added information about past studies. line 247-252

Though the model of the Hedonic price, widely used to determine implicit prices in food and beverage sector, most recent studies are focused on wine, [76-78] oil [79], diary, egg and meat [80-84] and fish [85,86]. At the best of our knowledge, no studies have explicitly analysed pulses using hedonic price functions. In the context of a relatively unexplored market, as the pulses case, the hedonic price model may give the pulse producer useful insights into the most important attributes to which they must pay attention to improve their profit.

DISCUSSION

There is a need for more discussion on how achieved findings can be connected to the previous conceptual background. Please link the discussion of the research problem with the highlighted gaps in the conceptual background. In fact, it is expected to use and criticise any relevant literature (in the literature review section) in order to reach the main purpose of the study to explain how the paper succeeded in enriching the development of the selected topic as well as to explore different views.

Thank you for your suggestion we added the sentences:

Line 377-382

 Our results suggest that market price may be significantly influenced by the exogenous effect of the origin-declaration on products. That is in line with others studies [56-61, 83] stressing on the positive attitude of Italian consumers to recognize a higher value to domestic and local products. That points to possible changes in market practices. The adoption of voluntary labelling option pointing out the origin from a specific territory may be effective and can help producers and sellers to differentiate the Italian product from the imported one

There are some typos, please make sure you proofread the paper

The manuscript has been carefully revised

THEORETICAL AND PRACTICAL IMPLICATIONS

To begin, the justification for your study from an industry/practical standpoint was lacking. This was a salient oversight on your part.

You should try to answer to the following questions: What kinds of objective evidence can you offer that would make industry leaders sit up and pay attention to your study? What makes this topic a big deal right now, and perhaps in the immediate future?

We added sentences from Line 409-419

A possible strategy to increase family consumption of local pulses, answering the need to support healthier and adherent to the patterns of the Mediterranean diet food consumptions [26,28,31] could be boosting local food presence in large scale distribution and discount outlet and in alternative food purchasing initiatives, such as box schemes or solidarity purchasing groups [65]. Suggestions to companies result from our insights: investing in voluntary labelling actions supplying consumers with a clear indication of place of origin and explaining the environmental and social benefits of local pulses; fruiting the synergy that can result from the indication of origin combined with organic certification [65]. Moreover, policy interventions supporting valorisation strategies of legume crops through their link with the territory may respond to the 2nd Sustainable Development Goals (SDGs) "Zero Hunger", harmonizing healthy food production and sustainable agriculture.

Best Regards

Round 2

Reviewer 2 Report

Dear author(s),

I'm happy with the revision made.

Best Regards